# Mapping of Outdoor Food and Beverage Advertising around Spanish Schools

**DOI:** 10.3390/nu14153167

**Published:** 2022-07-31

**Authors:** Ruben Martin-Payo, María del Rosario González-Moradas, Juan Iturrate-Bobes, Alejandro Fernández-Sutil, Rafael Cofiño, María del Mar Fernandez-Alvarez

**Affiliations:** 1PRECAM Research Group, ISPA Asturias-Spain, Faculty of Medicine and Health Sciences, University of Oviedo, 33006 Oviedo, Spain; fernandezmar@uniovi.es; 2Faculty of Mine Engineering, University of Oviedo, 33003 Oviedo, Spain; mrgmoradas@uniovi.es; 3Faculty of Medicine and Health Sciences, University of Oviedo, 33006 Oviedo, Spain; UO270856@uniovi.es (J.I.-B.); UO270026@uniovi.es (A.F.-S.); 4Consejería de Salud del Principado de Asturias, 33003 Oviedo, Spain; rafael.cofinofernandez@asturias.org

**Keywords:** advertising, food, beverage, food advertising, food environment, schools

## Abstract

Overweight and obesity rates have increased worldwide in the last decades. The marketing strategies of food considered to be unhealthy significantly exacerbate the childhood obesity dilemma. Studies typically analyze the content of advertisement in television, movies, or social media, but there is a gap in the assessment of the real-life promotion of food and beverages around the schools. The primary aim of the study was to assess the products advertised around public and concerted schools in three cities in the north of Spain, and to categorize them as healthy (core) or unhealthy (discretionary). The secondary aim was to describe the types of food and beverages in advertisements, as well as to determine the density of core and discretionary product advertisements. A cross-sectional descriptive study was carried out between September and December 2021. The units of analysis were outdoor food and beverage advertisements that were located around public and concerted schools of three cities in the north of Spain. We found 104 schools that met the criteria. We identified 6469 products advertised, 35.1% core and 61.2% discretionary, observing significant differences (*p* < 0.001) among the cities. Fruit (core) and alcohol (discretionary) were the most heavily advertised products. In conclusion, children attending schools located in the assessed cities are currently exposed to a significant amount of discretionary product advertisement, a situation that should be regulated.

## 1. Introduction

Overweight and obesity rates have increased worldwide in the last decades [1], so much so that childhood obesity has become one of the biggest international health challenges [2]. Data from the ALADINO study carried out in Spain in 2019 show that 23.3% of primary school students in Spain are overweight and 17.3% are obese [3,4]. According to the ESNUPI-AS study, the prevalence of children who were overweight and obese was 33.3%, with a higher rate in children aged 7–14 years, in the region where the study was carried out [5].

García-Solano et al. [4] suggested that to improve the current statistics, it is necessary to address the factors that influence childhood obesity, especially dietary factors. The results of the SENDO project in Spain indicated that there is still much to address, with only 74.4% of the children participating in the study showing a moderate adherence to the Mediterranean Diet [6]. Similar numbers were observed in a recent study that was conducted in Asturias, where only 54.8% of the participants had a medium adherence and 8.9% had low adherence to this diet [7].

While López-Sobaler et al. demonstrated that the promotion of healthy lifestyles at home is essential [8], these lifestyles should also be promoted in relevant environments for children, such as schools, as set out in Law 17/2011 of 5 July on food safety and nutrition [9]. Article 40 of this legal regulation includes several special measures to be implemented in schools: briefly, the law prevents the marketing and sale of unhealthy foods and beverages in schools.

However, there are no regulations in Spain that specifically address marketing around schools, and according to the literature reviewed, this marketing has a clear influence on children’s intake patterns [10,11,12]. A metanalysis by Sadeghirad et al. [13] showed that children exposed to marketing campaigns for unhealthy food significantly increased their intake of these products immediately after or very early after being exposed to said advertisements. Ponce-Blandon et al. [14] suggested that the effect is immediate and significant, especially in cases of wide brand awareness. The World Health Organization (WHO) has also shown that the promotional strategies employed in the generalized and persuasive marketing of unhealthy food significantly exacerbate childhood obesity [15].

Despite the evidence, children’s exposure to the commercialization of unhealthy food remains an unresolved issue. Hence, its situation constitutes an obstacle in the fight to end childhood obesity, especially in a region where the rates can be improved [5].

Although Spain has a specific regulation related to the marketing of food and beverages in schools [9], which has contributed to establishing food promotion campaigns in schools, no regulations have been found that regulate the advertising in school environments. This could be due to a lack of research demonstrating the problem. This research could highlight the need for developing health promotion strategies that minimize the exposure of minors to unhealthy products.

We conducted the present study after identifying a gap in the assessment of the real-life promotion of food and beverages around the schools in the Principality of Asturias. The primary aim of the study was to assess the products advertised around public and concerted schools in three cities in the north of Spain, and to categorize the products as healthy or unhealthy. The secondary aim was to describe the types of food and beverages in commercials, as well as to determine the core and discretionary product density.

## 2. Materials and Methods

### 2.1. Design

A cross-sectional descriptive study was carried out between September and December 2021 in three cities in the north of Spain, Oviedo, Gijón, and Avilés, in the Principality of Asturias.

### 2.2. Unit of Analysis

The Principality of Asturias is an autonomous community in the north of Spain, with approximately 1 million inhabitants distributed across 10,603 km^2^, and a mean population density of only 96 inhabitants/km^2^. It is characterized by an ageing, geographically dispersed, and rural population. However, most of the children reside in urban centers. Therefore, we examined all public or concerted schools situated in areas with more than 50,000 inhabitants and with a population density ≥ 1000 inhabitants/km^2^. The cities of Oviedo, Gijón, and Avilés met these criteria.

The units analyzed were all outdoor food and beverage advertisements that were located within a 500 m radius of all public and concerted schools for children aged 3 to 12 years old within the 3 cities. We include those advertisements in the streets or, if they belonged to an establishment, those located outside of the establishment and directed towards the street. The advertisements that could be seen from the street but were located inside the establishments were not included.

A 500 m radius was drawn around each school based on the coordinates obtained from Google Maps (Mountain View, California, Estados Unidos), following the methodology previously employed by other authors [16,17,18]. In addition, considering that the assignment of schools to minors in Spain is carried out according to their place of residence, this radius was defined as the area that includes both the minors’ school and place of residence, and therefore spanned the area of influence. These maps were stored in a digital form throughout the data collection.

### 2.3. Data Collection

We classified the products included in the advertisement, following the categories defined by Charlton et al. [19] and the Nutrient Profile Model [20], considering oils as a core category (classification included in in Appendix A). Therefore, products advertised were coded as major category (core/healthy/nutritious food and beverages; discretionary/unhealthy/ foods and drinks that provide nutrients that the body does not need; miscellaneous) and minor category (food/drink subtypes included in the major categories), and the density of core/discretionary advertisements around the school (within 500 m) was calculated. Core products correspond to those suggested by the Spanish Society of Community Nutrition to be eaten daily, weekly, or at each main meal, and discretionary products are those where consumption should be optional, occasional, and moderate [21].

We performed the data collection and analysis of the advertisements within a 10 week period to reduce the seasonal fluctuations that characterize advertisements. Before the data collection, a pilot study was conducted in streets that were not assessed in the subsequent study, with the aim of identifying the differences in our criteria, and thus avoiding heterogeneity in the measurements. Two researchers (AFS and JIB) oversaw the early data collection. Both researchers visited schools—AFS in Oviedo and Avilés, JIB for those in Gijón—and the streets included within the previously identified radius. During this process, they filled out a digital form detailing the basic information of each advertisement (geolocation, placement, and characteristics of the advertised product). They also took pictures of the respective advertisements. Subsequently, two researchers (RMP and MFA) were randomly assigned the results of the early data collection, and both researchers verified the information from the forms and the pictures, observing no differences between the data from the forms and the pictures.

Finally, the products advertised were coded using major and minor categories, and the density of core and discretionary products was estimated in each city, as well as the distance between each school and its nearest core product advertisement [19] (Appendix A).

The number of products advertised, per surface unit, within a 500 m radius of each school was calculated. Buffers were created by drawing a 500 m radius. The total surface area was estimated to be 785,398 m^2^ (surface area *S* = *π* × Radius^2^). The cutting point and scores for core and discretionary products were calculated for each buffer (*Numpoints*). The results are presented as density, calculated using ((*Numpoints*/785,398) × 10,000). The result is n × 10^−4^ (points/m^2^) (data in Appendix A). The analysis results were classified in 5 value groups (very low, low, average, high, and very high) and associated with schools to create graphic representations. 

### 2.4. Data Analysis

A descriptive analysis of the variables was performed on the mean (interquartile range (IQR)) and percentage, according to the nature of the variable. The Kolmogorov–Smirnov test was used to determine the normality of the variable distribution. A chi-square test was conducted to test the differences in core, discretionary, and miscellaneous products between the cities. The threshold for significance was *p* < 0.05. The differences regarding the distance from the nearest core product advertised were calculated using the Kruskal–Wallis test, with respect to each city. Data were analyzed with the software package IBM SPSS for Windows, Version 27 (SPSS Inc., Chicago, IL, USA).

A geographic information system (GIS) was created to perform a spatial analysis using QGIS 3.16.7-Hannover software. The reference system chosen for the GIS was ETRS 89, as it is official in Spain and is compatible with WGS 84. The system incorporated: (i) georeferenced information from the field research, that is, the schools and products advertised. Links to the spreadsheets were established, and point layers were created using longitude and latitude; (ii) WFS connection to SITPA-IDEAS (the geographical information system at the Principality of Asturias), the use of layers for municipalities and schools; (iii) WMTS connection to the orthophotos of the National Plan for Aerial Orthography (PNOA in Spanish) of the National Geographical Institute.

## 3. Results

The total number of schools assessed was 104, with 46.1% (n = 48) in Gijón, 33.7% (n = 35) in Oviedo, and 20.2% (n = 21) in Avilés. The total number of advertised products was 6473, with 53.1% in Gijón (n = 3439), 36.5% in Oviedo (n = 2360), and 10.4% in Avilés (n = 674).

### 3.1. Categorization of Products Advertised as Core, Discretionary, and Miscellaneous

Most of the products advertised belonged to the discretionary category. Significant differences were observed in both the core and discretionary categories among the different cities (Table 1).

### 3.2. Characteristics of the Products Advertised According to Type of Product

Fruit stood out as the most advertised core product and alcohol as the most advertised discretionary product, according to the analysis of minor food categories (Table 2).

### 3.3. Core and Discretionary Product Density

A higher density of discretionary versus core products advertised was observed in all cities. Oviedo presented the schools with the highest number of core products advertised observed in the school surroundings, while Avilés presented the opposite (Appendix A). Gijón presented the highest number of discretionary products advertised, followed by Oviedo, then Avilés, as seen in Appendix A. The distribution of both core and discretionary products advertised was higher around schools located in the city center than in those located in the outskirts. Avilés was the exception, where there was a more homogeneous distribution (Appendix A; each density point surrounds a school).

### 3.4. Distance from Schools to Nearest Core Products Advertised

The median distance from each school to the nearest core product advertised was calculated. Taking into consideration all schools regardless of the city, the median was 342.1 m (IQR = 589.5–198.9). In Gijón, 17 schools were less than 200 m away from the nearest core product advertised, 9 in Oviedo, and 0 in Avilés. The median was significantly lower in Oviedo, where it was 302.4 m (IQR = 496.8–185.0). The median in Gijon was 331.1 m (IQR = 582.9–166.3), and in Avilés, it was 455.3 m (805.9–313.9) (*p* = 0.026).

## 4. Discussion

The results of the present study indicate that the percentage of advertisements of unhealthy foods and beverages near schools in the cities of Gijón, Oviedo, and Avilés is significantly higher than that of healthy products advertised and, therefore, is a suboptimal circumstance. 

There is no perfect system for classifying foods into healthy and unhealthy. Authors of previous studies have used classification systems. For example, Parnell et al. [16] used the Australian Dietary Guidelines, Trapp et al. [17] preferred the Australian Guide to Healthy Eating food categories, and Dia et al. [22] chose the World Health Organization Nutrient Profile Report and Nutrient Profile Model. Although these authors did not indicate the reasons why these classification systems were selected, their choices seem to be related to their geographical and cultural context. We followed these criteria to use a food classification system developed to align with the three previous references: the categories defined by Charlton et al. [19], the World Health Organization Nutrient Profile Model [20], and the Spanish Society of Community Nutrition Dietary Guidelines for the Spanish Population [21].

Generally, the data observed are consistent with those of previous studies, with a greater presence of products advertised that are considered discretionary, regardless of the city [16,17,22,23]. Furthermore, a higher density for both core and discretionary products advertised was observed in the city centers versus the outskirts. This type of distribution was previously observed by other researchers, who emphasized a decreased density of advertisements when moving away from the city center to the outskirts [22]. Avilés was the exception, but the homogeneous distribution of this city may be related to its smaller size.

Previous studies analyzing the content of advertisement in other media, for example, in television, movies, or social media, also observed that advertisements favor unhealthy food or beverages versus healthy products [24,25,26,27,28].

Therefore, there is a marked tendency toward the advertisement of unhealthy products, which is harmful for children’s health. The evidence shows that the marketing of unhealthy foods and beverages leads to an increased intake and preference, especially of foods and beverages high in calories and low in nutrients [13,29], and it is intrinsically linked to potential weight gain [30].

From a purely economic perspective, it is not surprising to think of children as a priority target audience, as they represent a significant percentage of the population [31]. Furthermore, it is important to remember that children are a target audience for food and drink marketers, especially children under 8 years old, whose capacity to recognize the persuasive intent of advertisement is limited, making the advertisement strategies highly effective against them [32]. Several authors agreed with this, and they noted that the intake of products from establishments near schools is significantly related [33,34]. While the present study focused on the analysis of the products advertised rather than the establishments, it is necessary to highlight that the presence of the advertisements themselves influence intake, as noted by Velazquez et al. [35]. We find it reasonable to believe that advertisements are sometimes placed in the establishment where the mentioned product is sold, therefore making it accessible to children.

Among the results, the high percentage of fruit and vegetables was noted as a positive finding. Given the effect of marketing on children, we believe that it is reasonable to think that fruit advertisements near schools may promote intake. The benefits are evident, which have been resulted in school interventions to promote fruit intake [36].

Conversely, alcohol, desserts, ice cream, and unhealthy ready-made meals are negative points. The findings support the results from previous studies where these products were more heavily promoted around schools [21,24,37]. The intake of alcohol is problematic in the region where this study was performed. According to the regional data, 12% of the adult population consumes alcohol on a daily basis versus 9% at the national level, with an average of 40 and 20 g for men and women, respectively [38]. Data from the last ESTUDES survey showed that 49.2% of the Asturian population aged between 14 and 18 years has suffered at least one alcohol intoxication in their life [39].

To prevent the development of noncommunicable diseases, it is essential to decrease the intake of sugars, red and processed meat, salt, and sugary beverages, and to instead increase the intake of fish, fruits, legumes, minimally processed wholegrains, non-starchy vegetables, nuts, and oils that are high in unsaturated fats [40]. To achieve this, several authors have highlighted that only those countries implementing legal regulations reduce the impact of the advertisement of unhealthy products that target children and, therefore, the intake of unhealthy foods and beverages in this population [41]. Several countries have implemented sets of regulations, such as taxes on unhealthy foods [40] and limiting their advertising [32].

In Spain, the PAOS Code [42] is as old as it is futile, as suggested by several publications [43,44]. This may be rooted in the weakness of the public health system in monitoring the code’s implementation. It is also important to highlight that establishing regulations by themselves is generally not enough, and it is essential to develop pedagogical practices, alliances, and compromises for all the people and populations potentially involved, such as the private sectors that are responsible for food or marketing. 

The Draft of the Royal Decree on food and drink advertisement targeting children, as the name implies, aims to regulate marketing aimed at children [45]. In article 2 of the Royal Decree, marketing aimed at children is defined as advertisements displayed on television channels (general or children’s) during children’s viewing times, cinemas, children’s magazines, webpages, apps, social networks, and video exchange sites [45]. Hence, despite the evidence of the impact of advertisements around schools, this type of marketing does not seem to be included in the new regulation. It is somewhat paradoxical, as the effectiveness of limiting unhealthy product advertisements around schools is well-known. According to Liu et al. [46], removing these advertisements within 400 m of schools would result in a 25% reduction in the exposure of children to unhealthy product advertisements.

### Strengths and Limitations

To the best of our knowledge, no published research has examined outdoor food and beverage advertising in Spain, this being one of the strengths of the present study. The results obtained could be considered by politicians and health authorities, not only in the Principality of Asturias, but also in other territories of Spain. The results allow for reflection on the impact of advertisements in the fight against childhood obesity, and could lead to the implementation of new regulations. Additionally, future studies may focus on analyzing the impact of these results on policies.

A limitation of this study was that only the three cities were considered, requiring more than 50,000 inhabitants and 1000 inhabitants/m^2^. Therefore, the results obtained could vary if other areas in the same region were considered. However, it should be noted that these areas were selected because most of the region’s child population resides in these cities, which should be taken into consideration, as a high percentage of this population could be exposed to the advertisements of unhealthy products.

## 5. Conclusions

The results of the present study demonstrate that children attending schools located in the cities assessed are exposed to a significant quantity of discretionary food and drink advertisements. This suggests the need to increase core product advertisements and decrease discretionary product advertisements. Intervention by politicians and health authorities by means of regulating advertisements around schools is justified by the findings of our research.

## Figures and Tables

**Table 1 nutrients-14-03167-t001:** Percentage of major categories products advertised around the schools, classified by city.

	Total	Gijón	Oviedo	Avilés	*p*
% Core	35.1 (2272)	34.0 ^a^ (1168)	38.0 ^b^ (896)	30.9 ^a^ (208)	<0.001
% Discretionary	61.2 (3961)	62.0 ^a^ (2134)	58.5 ^b^ (1380)	66.2 ^a^ (447)	<0.001
% Miscellaneous	3.7 (240)	4.0 (137)	3.5 (84)	2.8 (19)	0.302

*p* < 0.001 between ^a^ and ^b^.

**Table 2 nutrients-14-03167-t002:** Percentage of type of minor foods and beverages around the schools classified by city.

	Total	Gijón	Oviedo	Avilés
**Core % (n)**				
Grains	2.8 (178)	2.8 (97)	2.9 (69)	1.7 (12)
Fruit	12.1 (780)	13.4 (459)	12.0 (283)	5.6 (38)
Vegetables	7.2 (463)	7.7 (266)	6.5 (153)	6.5 (44)
Dairy and alternatives	1.0 (67)	0.6 (21)	1.7 (40)	0.9 (6)
Meat and alternatives	11.2 (727)	8.7 (298)	13.8 (326)	15.3 (103)
Oils	0.7 (43)	0.5 (18)	0.8 (20)	0.7 (5)
Water	0.2 (14)	0.3 (9)	0.2 (5)	0 (0)
**Discretionary % (n)**				
Processed meats and jam	8.1 (524)	7.0 (239)	7.0 (165)	17.8 (120)
Chips and savory snacks	1.1 (73)	1.2 (41)	0.9 (21)	1.6 (11)
Fruit beverages	0.3 (20)	0.3 (12)	0.3 (7)	0.1 (1)
Other beverages	1.5 (95)	1.7 (60)	1.2 (28)	1.0 (7)
Desserts and Ice cream	13.4 (869)	15.4 (531)	12.1 (287)	7.6 (51)
Chocolate and candy	2.4 (155)	3.2 (112)	1.2 (29)	2.1 (14)
Unhealthy ready meals, salt, and fat products	12.3 (800)	16.6 (570)	7.1 (168)	9.2 (62)
Soft and energy beverages	6.4 (415)	6.2 (214)	6.7 (158)	6.4 (43)
Alcohol	15.6 (1010)	10.3 (355)	21.9 (517)	20.5 (138)
**Miscellaneous/Other % (n)**				
Recipe additions	0.2 (13)	0.2 (6)	0.2 (5)	0.3 (2)
Tea and coffee	3.5 (227)	3.8 (131)	3.3 (79)	2.5 (17)

## Data Availability

Not applicable.

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
