# Peer review of "Mapping of Outdoor Food and Beverage Advertising around Spanish Schools"

_nutrients, 2022, doi:10.3390/nu14153167_

Round 1

Reviewer 1 Report

The study provides insight into food advertising targeting children that may impact the high childhood obesity rates in ret region of Asturias in Northern Spain. While the results are outlined in Tables 1 and 2, the clarity of language used to describe the study aims, design and results must be improved to help the reader. 

Major points:

1.     In the abstract (and thereafter throughout the manuscript text), advertised products are referred to as “products” which is confusing and incorrect. Please clarify that you are describing advertisements and call them “products advertised”.

2.     In the abstract, what constitutes “core” and “discretionary” products must be briefly defined in the context of the study and then further outlined in the methods/data collection section.

3.     For figures 1 and 2, readability is poor, and the figure text is lacking. As a reader, it was necessary to study the figures quite a bit to figure out what is actually represented. The fact that each oval represents a school is not described in the figure text and should be. This represents the density of core or discretionary product advertisements and not products in in a 500m radius of each school---that should be stated. The font size is very tiny and not well resolved and must be corrected. A, B, and C for each figure would also help.

4.     The conclusion paragraph is poorly wording and requires editing—“intervention…seems convenient” is not a strong or convincing statement.

Minor points:

5.     Paragraph #1 on page 2 (lines 46-50) is unclear due to awkward wording.

6.     Table 1 description should be edited to read “…and their distribution near schools in each city.”

7.     In lines 192 and 198, “inferior” and “superior” are used to describe median distance and the percentage of unhealthy foods advertised. The word choice doesn’t make sense.

Author Response

Major points:

  1. In the abstract (and thereafter throughout the manuscript text), advertised products are referred to as “products” which is confusing and incorrect. Please clarify that you are describing advertisements and call them “products advertised”.

Thanks for the comments. We made changes in the text.

  1. In the abstract, what constitutes “core” and “discretionary” products must be briefly defined in the context of the study and then further outlined in the methods/data collection section.

A clarification and brief explanation have been included in both sections, as previous authors have previously done.

  1. For figures 1 and 2, readability is poor, and the figure text is lacking. As a reader, it was necessary to study the figures quite a bit to figure out what is actually represented. The fact that each oval represents a school is not described in the figure text and should be. This represents the density of core or discretionary product advertisements and not products in in a 500m radius of each school---that should be stated. The font size is very tiny and not well resolved and must be corrected. A, B, and C for each figure would also help.

To improve the quality of the figures we attach Supplementary materials (file 1).

  1. The conclusion paragraph is poorly wording and requires editing—“intervention…seems convenient” is not a strong or convincing statement.

The conclusion has been edited to include stronger wording based on the results obtained.

Minor points:

  1. Paragraph #1 on page 2 (lines 46-50) is unclear due to awkward wording.

Wording has been modified to make it more understandable.

  1. Table 1 description should be edited to read “…and their distribution near schools in each city.”

Wording has been modified following this recommendation.

  1. In lines 192 and 198, “inferior” and “superior” are used to describe median distance and the percentage of unhealthy foods advertised. The word choice doesn’t make sense.

The text has been edited.

Reviewer 2 Report

Methodology and reporting must be improved. Based on current version, this study would not be able to be replicated due to the lack of details provided in section. Authors need to elaborate extensively on this section of the manuscript in particular, as it will leave readers with several questions and the bleakness of the described study design and associated procedures could lead readers to question the study validity. 

Additional context needs to be provided in the introduction to describe the current community/city this study takes place -- what value does this study have to this specific geographic area? 

Methodology describing how food products were defined as "healthy" and "unhealthy" should be included. Please elaborate on definitions and descriptions of what major and minor food categories consist of in methods section. Please also define all inclusion and exclusion criteria for each of the food categories listed in Table 2 (i.e. does grains include only whole grains? Is fruit only fresh fruit or is canned/frozen included? etc.)

Consider including Appendix or supplementary material with listed food and beverage items that fell within each category.

Please include in methods section further explanation and justification as to why a 500 mi radius was selected by study authors. 

Please include full calculations for density values for all areas in results section. 

Have authors considered the results in the context of the different level of schools i.e. elementary, middle, high school. Was this controlled for in analysis and investigated? Further, were daycare centers included in the school count? Please indicate in methods answers to these questions. 

The discussion provides some valuable points, such as including the economic implications. This should be introduced and elaborated on in the introduction, expanded upon in the discussion and connecting the results of the study to this principle in addition to the health implications. The paragraph discussing policy changes should be the crux of the discussion and elaborated on. 

In general, the manuscript in its entirety requires "more" to share why this study is novel. Authors are on target to address an important research question, but in order to be suitable for manuscript format additional context and scope should be considered and included. 

Author Response

Methodology and reporting must be improved. Based on current version, this study would not be able to be replicated due to the lack of details provided in section. Authors need to elaborate extensively on this section of the manuscript in particular, as it will leave readers with several questions and the bleakness of the described study design and associated procedures could lead readers to question the study validity.

Methodology was rewritten to make it more understandable and replicable.

Additional context needs to be provided in the introduction to describe the current community/city this study takes place -- what value does this study have to this specific geographic area?

Authors consider that it´s explained in methodology section (The Principality of Asturias is an autonomous community in the north of Spain, with approximately 1 million inhabitants distributed in 10,603 km2 and a mean population density of only 96 inhabitants/km2. It is characterized by an ageing, geographically dispersed, and rural population. However, most children reside in urban centers. Therefore, all public or concerted schools, situated in areas with more than 50,000 in-habitants and with a population density ≥ 1,000 inhabitants/km2. The cities of Oviedo, Gijón, and Avilés met these criteria), section 4.1 (To our knowledge, no published research has examined outdoor food and bever-age advertising in Spain, being this one of the strengths of the present study. The results obtained could be considered by politicians and health authorities, not only in the Principality of Asturias, but also other territories in Spain, to reflect on the impact of advertisements in the fight against childhood obesity, and lead to the implementation of regulations) and conclusion.

As stated in the article, we are sure that the results presented in this document will be echoed in the press and other media and we hope that they will promote changes in the environments analyzed. In fact, this is one of the objectives of public health research, to motivate policies that contribute to improving the health of the population. The components of the research group that have developed this study already have previous experiences in this regard. For example, as a result of similar analyses, we promoted the change from unhealthy to healthy vending machines in hospitals in our region.

Methodology describing how food products were defined as "healthy" and "unhealthy" should be included. Please elaborate on definitions and descriptions of what major and minor food categories consist of in methods section. Please also define all inclusion and exclusion criteria for each of the food categories listed in Table 2 (i.e. does grains include only whole grains? Is fruit only fresh fruit or is canned/frozen included? etc.)

As indicated in methodology section Products included in the advertisement were coded following the categories defined by Charlton et al. (19). It appears in Table A. If the reviewer checked the table, he/she can find answer to her/his doubts about the products classification (highlighted), as readers could do too.

We consider that it is better to just include the reference so not to make the paper too long especially, as it can be checked by readers. Nonetheless, if the reviewer or the editor considered it necessary, we could include in our paper a table similar to A or as indicated in the next comment as supplementary material.

Table A. Codes defined by Charlton et al. (19)

Category

Included

Core

Fruits

Fresh, canned, dried, fruit juices, pre-packaged fruit mixes

Vegetables

Fresh, canned (baked beans), frozen, pickled vegetables, pre-packaged vegetable mixes, vegetable only soup (tomato/pumpkin soup), tomato puree and pastes, vegetable juices, olives, sundried tomatoes

Dairy and alternatives

Milk (fresh and long-life), soy and other milk alternatives, all yoghurt (added fruit & full fat), cheese, cream cheese, cultured milk, custards, breakfast cereal beverages

Meat and alternatives

Fresh/frozen/roasted, beef, lamb, pork, chicken, veal, mince, crumbed raw fillets, fish, canned fish (tuna), smoked fish, seafood products, tofu, eggs

Grains

Rice, lentils, chickpeas, flour, pasta (fresh and dried), oats, bread (fruit breads), bread rolls, crumpets, English muffins, flat breads (pita), instant noodles, canned spaghetti, breakfast cereals (< 30 g/100 g sugar or < 35 g/100 g sugar if includes fruit), savoury biscuits (energy < 1800 KJ/100 g), rye, corn and rice based biscuits, raw nuts and seeds, unsalted nuts, plain popcorn

Water

Unflavoured mineral water, sparkling water

Fruit juice

100% fruit juice

Discretionary

Fats and oils

Butters, margarines, alternative spreads, oils

Processed meat

Sausages, rissoles, hamburgers, bacon, processed delicatessen meats (salami, ham), dried meats

Jams

Confectionary & chocolate

Blocks, bars, chocolate coated products, lollies, chocolate toppings

Chips

Potato crisps, corn chips, other crisps

Desserts & ice-creams

Cake, cake mixes, sweet biscuits, slices, scones, canned fruit in syrup, frozen yoghurt, sweet breads (buns, scrolls), pastries (croissant), doughnuts, icy poles

Unhealthy ready meals

Chicken nuggets, garlic bread, pizza, dumplings, spring rolls, crumbed/fried/battered meats, frozen or ready to eat chips and wedges, ready to eat meat products in sauce, meat pies, sausage rolls, ready to eat burgers, kiev, schnitzel

Other snacks

Breakfast cereals (> 30 g/100 g sugar or > 35 g/100 g sugar if includes fruit), savoury biscuits (energy > 1800 KJ/100 g), dips, salted roasted nuts, muesli bars, snack bars, salted and flavoured popcorn

Soft drinks

Full sugar soft drinks

Soft drinks diet

Diet, sugar-free soft drinks

Energy drinks

energy drinks

Fruit drink

Fruit flavoured drinks

Cordial

Pre-prepared syrups and concentrate

Other drinks

Flavoured milks, flavoured mineral waters, vitamin waters, sport drinks, electrolyte drinks

Discretionary-other

Cream, pastry sheets, sugar, condensed milk, icing sugar

Alcohol

Beers, wines, spirits, alcoholic mixers

Other

Products that could not be classified into any other category, tea, coffee, milk powder, natural sweeteners, chewing gum, mixed dishes (ingredients from multiple categories that did not fall predominately under one sub-category), ready to eat and frozen meals that could not be classified as unhealthy (weight control meals), stock powder and liquid, salad dressing, sauces, vinegar, salt, breakfast spreads, mayonnaise, herbs and spices, protein powders, infant food products, mixed salads with dressing

Please include in methods section further explanation and justification as to why a 500 mi radius was selected by study authors.

Thanks for the suggestion. We clarify this aspect in section 2.2. “A 500-meter radius, based on the coordinates from Google Maps, was drawn around each school, following the methodology previously employed by other authors [16-18]. In addition, considering that the assignment of schools to minors in Spain is carried out according to their place of residence, this radius was considered to be the area that includes both the minors' residence and school, and therefore the area of influence.”

Please include full calculations for density values for all areas in results section.

Text changed in the manuscript. I attach here the values but I think is totally unnecessary to include this information in the manuscript.

Productos “Core”

Gijón

School code

Density

33005350

0,43

33006342

0,00

33028601

0,00

33005362

1,18

33020545

1,20

33020557

0,56

33022554

1,04

33022578

1,06

33020570

1,03

33022104

0,55

33006445

0,14

33021914

0,29

33023601

1,99

33022566

0,29

33020314

0,15

33005404

1,08

33006007

2,42

33021926

1,74

33005957

1,97

33005969

2,38

33006081

0,20

33005428

2,98

33021872

0,04

33006494

0,04

33021653

2,72

33019701

2,66

33028982

1,66

33022335

0,38

33006093

0,33

33005945

0,00

33006287

0,27

33005891

2,16

33005982

2,42

33005933

1,77

33027874

0,05

33005398

2,83

33005908

3,08

33005881

1,85

33024289

2,57

33005911

0,36

33005337

0,17

33005878

0,51

33021938

0,20

33006044

0,00

33021665

0,04

33006184

0,01

33005210

0,00

Oviedo

School code

Density

33013085

0,04

33012721

0,00

33012731

0,00

33012664

4,07

33021781

3,11

33022581

1,69

33012378

1,11

33013103

0,14

33013152

1,41

33012639

2,27

33012688

0,37

33012457

1,53

33012755

0,25

33012408

0,98

33012767

0,27

33012652

0,94

33012640

0,61

33012421

1,86

33012411

1,16

33022281

0,92

33012627

1,03

33022347

1,86

33023613

1,12

33012743

0,89

33012573

0,99

33013097

0,45

33012691

0,95

33012676

0,47

33012433

0,42

33020909

0,24

33021471

0,52

33012044

0,85

33012536

0,18

33028672

0,05

33028064

0,13

Avilés

School code

Density

33001216

0,00

33001472

0,00

33001319

0,67

33022098

0,62

33001022

0,00

33001344

0,43

33001046

0,00

33001137

0,89

33001228

0,00

33020132

0,51

33001034

0,65

33001435

1,08

33001186

0,22

33001095

0,69

33022086

0,46

33001149

0,33

33001174

0,20

33001150

0,27

33022256

0,08

33001460

0,20

33001231

0,20

Discretionary

Gijón

School code

Density

33005350

0,45

33006342

0,25

33028601

0,06

33005362

0,78

33020545

1,03

33020557

0,50

33022554

1,59

33022578

1,57

33020570

1,68

33022104

0,60

33006445

0,24

33021914

0,79

33023601

5,90

33022566

0,79

33020314

1,07

33005404

3,82

33006007

3,11

33021926

4,39

33005957

4,23

33005969

2,60

33006081

2,30

33005428

4,21

33021872

0,15

33006494

0,15

33021653

3,91

33019701

3,60

33028982

6,47

33022335

5,53

33006093

5,09

33005945

0,00

33006287

2,95

33005891

1,91

33005982

1,87

33005933

3,21

33027874

0,13

33005398

3,30

33005908

3,25

33005881

3,31

33024289

2,98

33005911

1,12

33005337

0,51

33005878

1,69

33021938

0,71

33006044

0,00

33021665

0,18

33006184

0,03

33005210

0,00

Oviedo

School code

Density

33021471

0,97

33012044

2,51

33012536

0,39

33028672

0,24

33028064

0,71

33013085

0,09

33012721

0,00

33012731

0,01

33012664

3,11

33021781

2,98

33022581

1,99

33012378

1,71

33013103

0,33

33013152

1,90

33012639

2,19

33012688

0,97

33012457

1,80

33012755

0,62

33012408

1,54

33012767

0,39

33012652

2,97

33012640

0,80

33012421

3,40

33012411

2,95

33022281

1,04

33012627

1,66

33022347

3,44

33023613

2,02

33012743

1,77

33012573

1,13

33013097

0,92

33012691

1,77

33012676

0,83

33012433

1,09

33020909

0,36

Avilés

School code

Density

33001216

0,00

33020132

1,52

33001034

0,97

33001435

1,09

33001186

0,81

33001095

1,15

33022086

1,06

33001149

0,33

33001174

0,48

33001150

0,52

33022256

0,32

33001472

0,00

33001460

0,27

33001231

0,22

33001319

1,34

33022098

1,52

33001022

0,06

33001344

1,01

33001046

0,06

33001137

1,58

33001228

0,09

Have authors considered the results in the context of the different level of schools i.e. elementary, middle, high school. Was this controlled for in analysis and investigated? Further, were daycare centers included in the school count? Please indicate in methods answers to these questions.

We included only elementary schools, which in Spain include students from 3 to 12 years old. We clarify this aspect in methodology section. We didn’t consider to additionally include people over 13 years. To include younger children is absolutely impossible due to the organization of the educational system. Nonetheless, thanks for the recommendation.

The discussion provides some valuable points, such as including the economic implications. This should be introduced and elaborated on in the introduction, expanded upon in the discussion and connecting the results of the study to this principle in addition to the health implications. The paragraph discussing policy changes should be the crux of the discussion and elaborated on.

We do not consider extending these aspects further because they are not the object of our analysis but rather consequences of the findings. Perhaps if the objective of the study focused on these aspects, i.e., analysis of the intentionality of companies or an analysis of the evolution of policies, we should focus more on this development. We consider that the expansion of the study suggested by the reviewer would be based on contributions from other authors and could divert the objective of our study.

However, we have included a modification in the implications. As we indicate above, the aim of public health research is to motivate changes in policies, so, “Future studies may focus on analyzing the impact of these results on policies”

In general, the manuscript in its entirety requires "more" to share why this study is novel. Authors are on target to address an important research question, but in order to be suitable for manuscript format additional context and scope should be considered and included.

This research is ABSOLUTELY novel in Spain, because, to our knowledge, there are no previous studies with the same aim. It could contribute to explain why there is no policies to regulate advertisings around schools. Even, the Draft of the new Royal Decree on food and drink advertisement targeting children, that, as the name implies, aims to regulate marketing aimed at children do not consider this advertisement. As indicated above, we are sure that our results will have a high impact in press and policies.

Reviewer 3 Report

Introduction
- The introduction doesn´t clearly the relevance of the study.
- The Literature Review can be more concise and avoid nominalizations.
- The introduction doesn´t clearly delineate the problem of the study.

Methods
- In my opinion, the methods needs to be better written.
- It is not clear how the selection of schools was made. Authors need to describe the characteristics of the sample. 
- It is necessary to better describe the data collection and the instruments used.

Results
- The tables are not self-explanatory and should be made clearer.
Discussion
-       The authors could describe if school food programs have defined nutritional standards.
-       The authors should point out the main conclusions of the study.

Author Response

Introduction

The introduction doesn´t clearly the relevance of the study. The introduction doesn´t clearly delineate the problem of the study.

As stated in the article, to our knowledge, there are no previous studies with the same aim. This could help explain why there are no policies to regulate advertisements around schools. As suggested Lord Kelvin, “what is not measured, cannot be improved”. So, these results iare relevant to promote changes in the environments analyzed. In fact, this is one of the objectives of public health research, to motivate policies that contribute to improving the health of the population. The components of the research group that have developed this study already have previous experiences in this regard. For example, as a result of similar analyses, we promoted the change from unhealthy to healthy vending machines in hospitals in our region.

The Literature Review can be more concise and avoid nominalizations.

We consider that literature is adequate, updated and pertinent, and nominalizations are totally accepted. We are not going to change these aspects.

Methods

In my opinion, the methods need to be better written. It is not clear how the selection of schools was made. Authors need to describe the characteristics of the sample. It is necessary to better describe the data collection and the instruments used.

Thanks for the suggestion. We have improved this section.

Results

The tables are not self-explanatory and should be made clearer.

We made a change in this sense.

Discussion

The authors could describe if school food programs have defined nutritional standards.

Sorry, but we think that this point has no sense since the study does not address food programs in schools.

The authors should point out the main conclusions of the study.

The conclusion has been edited to include stronger wording based on the results obtained.

Round 2

Reviewer 2 Report

Original reviewer suggestion: Methodology describing how food products were defined as "healthy" and "unhealthy" should be included. Please elaborate on definitions and descriptions of what major and minor food categories consist of in methods section. Please also define all inclusion and exclusion criteria for each of the food categories listed in Table 2 (i.e. does grains include only whole grains? Is fruit only fresh fruit or is canned/frozen included? etc.)

Author response: As indicated in methodology section Products included in the advertisement were coded following the categories defined by Charlton et al. (19). It appears in Table A. If the reviewer checked the table, he/she can find answer to her/his doubts about the products classification (highlighted), as readers could do too.

We consider that it is better to just include the reference so not to make the paper too long especially, as it can be checked by readers. Nonetheless, if the reviewer or the editor considered it necessary, we could include in our paper a table similar to A or as indicated in the next comment as supplementary material.

Reviewer response: As originally indicated, this feedback was provided because despite reference to the table and citation, what was provided was inadequate -- hence the suggestion to include and elaborate. Table A should absolutely be included in the Supplementary Material. 

Original reviewer suggestion: Please include full calculations for density values for all areas in results section.

Author response: Text changed in the manuscript. I attach here the values but I think is totally unnecessary to include this information in the manuscript.

Reviewer response: Including the calculations is absolutely necessary, thank you for providing those in the text. Unsure what the subsequent table means in response as detail and response was limited?

Further, it is inappropriate to assume and accuse a reviewer of not reviewing manuscript content, as providing constructive feedback is a vital and necessary part of the peer review process and should be graciously received by authors. The feedback provided to authors on the originally submitted manuscript is unbiased, which authors would be wise to remember. Despite your opinion of the caliber of manuscript submitted, there is always room for improvement and often details are overlooked when presenting research in an academic format. If authors have an aversion to constructive feedback, perhaps they should consider submitting their findings to a lower caliber journal or nonacademic outlet. It is my hope that your future responses to reviewers will be more professional and organized, as this response by authors was lazy, inadequate, and unbecoming. 

Author Response

Reviewer response: As originally indicated, this feedback was provided because despite reference to the table and citation, what was provided was inadequate -- hence the suggestion to include and elaborate. Table A should absolutely be included in the Supplementary Material.

Thanks for the recommendation. I agree with you that the reference is insufficient since we adapt to our context. Supplementary file 3 include the table

Reviewer response: Including the calculations is absolutely necessary, thank you for providing those in the text. Unsure what the subsequent table means in response as detail and response was limited?

Thanks for the recommendation. Supplementary file 2 include the data

Further, it is inappropriate to assume and accuse a reviewer of not reviewing manuscript content, as providing constructive feedback is a vital and necessary part of the peer review process and should be graciously received by authors. The feedback provided to authors on the originally submitted manuscript is unbiased, which authors would be wise to remember. Despite your opinion of the caliber of manuscript submitted, there is always room for improvement and often details are overlooked when presenting research in an academic format. If authors have an aversion to constructive feedback, perhaps they should consider submitting their findings to a lower caliber journal or nonacademic outlet. It is my hope that your future responses to reviewers will be more professional and organized, as this response by authors was lazy, inadequate, and unbecoming.

Sorry if you feel damaged. I totally agree with you that a good peer review process can improve the quality of the papers. I didn’t assume or accuse you of not reviewing, oppositely, I considered your comments useful as I wrote to the editor. In any case, if you read the comments of other reviewer maybe you also agree with me. So, we have no aversion to constructive feedback.

Why I answer shortly some recommendations was because of 1 reason: I don’t usually repeat data in the papers, for example, density in map and table. Even, previous reviewers in other publication recommended us the opposite: include data or maps. So, as I indicate in my reply, if you consider that it can improve the paper I accept and, as you can check, the table is added (supplementary file 3).

Finally, I consider inappropriate your final comments. We suppose that those are based in your perception that we accuse you to not review the manuscript. So, we are not going taken into account.